# Diabetes mellitus and tuberculosis, a systematic review and meta-analysis with sensitivity analysis for studies comparable for confounders

Joseph Rodrigue Foe-Essomba[1,2,3], Sebastien Kenmoe[4], Serges Tchatchouang[5], Jean Thierry Ebogo-Belobo[6], Donatien Serge Mbaga[7], Cyprien Kengne-Ndé[8], Gadji Mahamat[7], Ginette Irma Kame-Ngasse[6], Efietngab Atembeh Noura[6], Chris Andre Mbongue Mikangue[7], Alfloditte Flore Feudjio[9], Jean Bosco Taya-Fokou[7], Sabine Aimee Touangnou-Chamda[7], Rachel Audrey Nayang-Mundo[10], Inès Nyebe[7], Jeannette Nina Magoudjou-Pekam[9], Jacqueline Félicité Yéngué[11], Larissa Gertrude Djukouo[9], Cynthia Paola Demeni Emoh[7], Hervé Raoul Tazokong[7], Arnol Bowo-Ngandji[7], Eric Lontchi-Yimagou[12], Afi Leslie Kaiyven[13], Valerie Flore Donkeng Donfack[3], Richard Njouom[4], Jean Claude Mbanya[12], Wilfred Fon Mbacham[14], Sara Eyangoh[3]*

1 Camdiagnostic, Ministry of Scientific Research and Innovation, Yaounde, Cameroon, 2 Faculty of Medicine and Biomedical Sciences, The University of Yaounde I, Yaounde, Cameroon, 3 Department of Mycobacteriology, Centre Pasteur of Cameroon, Yaounde, Cameroon, 4 Virology Department, Centre Pasteur of Cameroon, Yaounde, Cameroon, 5 Bacteriology Department, Centre Pasteur of Cameroon, Yaounde, Cameroon, 6 Medical Research Centre, Institute of Medical Research and Medicinal Plants Studies, Yaounde, Cameroon, 7 Department of Microbiology, The University of Yaounde I, Yaounde, Cameroon, 8 Evaluation and Research Unit, National AIDS Control Committee, Yaounde, Cameroon, 9 Department of Biochemistry, The University of Yaounde I, Yaounde, Cameroon, 10 Department of Microbiology, Protestant University of Central Africa, Yaounde, Cameroon, 11 Department of Animals Biology and Physiology, The University of Yaounde I, Yaounde, Cameroon, 12 Laboratory for Molecular Medicine and Metabolism, The University of Yaounde I, Yaounde, Cameroon, 13 Institute of Biomedical and Clinical Research, University of Exeter, Exeter, United Kingdom, 14 The Biotechnology Centre, The University of Yaounde I, Yaounde, Cameroon

* eyangoh@pasteur-yaounde.org

## Abstract

### Introduction

Meta-analyses conducted so far on the association between diabetes mellitus (DM) and the tuberculosis (TB) development risk did not sufficiently take confounders into account in their estimates. The objective of this systematic review was to determine whether DM is associated with an increased risk of developing TB with a sensitivity analyses incorporating a wider range of confounders including age, gender, alcohol consumption, smoke exposure, and other comorbidities.

### Methods

Pubmed, Embase, Web of Science and Global Index Medicus were queried from inception until October 2020. Without any restriction to time of study, geographical location, and DM and TB diagnosis approaches, all observational studies that presented data for associations between DM and TB were included. Studies with no abstract or complete text, duplicates,

**Data Availability Statement:** All relevant data are within the paper and its S1–S3 Figs, S1–S8 Tables files.

**Funding:** - Initials: SK - Grant number: VARIAFRICA-TMA2019PF-2705 - URL: http://www.edctp.org/projects-2/edctp2-projects/edctp-preparatory-fellowships-2019/ - The funders had no role in study design, data collection and analysis, decision to publish, or preparation of the manuscript. This project is part of the EDCTP2 programme supported by the European Union under grant agreement TMA2019PF-2705.

**Competing interests:** The authors have declared that no competing interests exist.

and studies with wrong designs (review, case report, case series, comment on an article, and editorial) or populations were excluded. The odds ratios (OR) and their 95% confidence intervals were estimated by a random-effect model.

## Results

The electronic and manual searches yielded 12,796 articles of which 47 were used in our study (23 case control, 14 cross-sectional and 10 cohort studies) involving 503,760 cases (DM or TB patients) and 3,596,845 controls. The size of the combined effect of TB risk in the presence of DM was OR = 2.3, 95% CI = [2.0–2.7], $I^2$ = 94.2%. This statistically significant association was maintained in cohort (OR = 2.0, CI 95% = [1.5–2.4], $I^2$ = 94.3%), case control (OR = 2.4, CI 95% = [2.0–2.9], $I^2$ = 93.0%) and cross-sectional studies (OR = 2.5, CI 95% = [1.8–3.5], $I^2$ = 95.2%). The association between DM and TB was also maintained in the sensitivity analysis including only studies with similar proportions of confounders between cases and controls. The substantial heterogeneity observed was mainly explained by the differences between geographic regions.

## Conclusions

DM is associated with an increased risk of developing latent and active TB. To further explore the role of DM in the development of TB, more investigations of the biological mechanisms by which DM increases the risk of TB are needed.

## Review registration

PROSPERO, CRD42021216815.

## Introduction

About 25% of the global population is infected with Mycobacterium tuberculosis (MTB) [1], including nearly 10 million new cases of active tuberculosis (TB) and 1.5 million deaths recorded each year [2]. These statistics have crowned TB as one of the leading causes of death from infectious diseases worldwide. MTB infections are more prevalent in developing regions of Southeast Asia (44%), Africa (25%) and the West Pacific (18%), with 2/3 of cases recorded in India, Indonesia, China, Philippines, Pakistan, Nigeria, Bangladesh and South Africa [2]. The International Diabetes Federation estimated that nearly half a billion people (about 10% of the global population) were living with diabetes mellitus (DM) each year, including more than 4 million deaths [3]. This incidence is predicted to increase by more than 10% by 2045, leading to about 700 million cases. The majority of people living with DM are registered in the urban areas of low-and middle-income countries where TB is also dominant. Five of 8 countries with the highest incidence of TB are among the 10 countries with the highest prevalence of DM [2, 3].

Compared to patients with TB only, patients with TB and DM are more likely to have more severe clinical pictures, greater infectivity, treatment failure for TB, relapses after recovery, and high mortality [4–8]. The global escalation of DM and TB epidemics is therefore detrimental and especially for low-resource countries where a very high proportion of DM remains undiagnosed or untreated due to poor resourced health systems [9, 10]. This high increase of DM patients in areas with high TB endemicity is of great concern to TB control efforts because

numerous studies have suggested that DM increases the risk of developing latent and active TB [11, 12]. Diabetes mellitus is indeed a disease that can alter the host's immunity and lead to increased susceptibility to several diseases including tuberculosis [13]. The association between DM and TB has been established in several systematic reviews including active TB [14, 15], latent TB [16] and multidrug-resistant TB [17, 18]. There are multiple confounding factors for the association between DM and TB, the main ones being: HIV infections [19, 20], undernutrition [21], smoking and alcoholism [22, 23]. Although all of these reviews have been devoted to the association between DM and TB, apart from adjusting analyses for age [14, 24], other major confounding factors such as HIV infection, alcohol or smoke exposure have received very little attention. In view of the increasing incidence of DM epidemic, further evidence of the association of DM and TB would be of crucial importance in the fight against the double DM-TB epidemic [25]. Furthering this knowledge could include implications such as the implementation of education, prevention, two-way early detection and co-management programs for MD and TB [26]. In this meta-analysis, including a sensitivity analysis with studies with similar proportions of confounders among cases and controls, we further assess the association between DM and TB.

## Methods

### Literature search

Preferred Reporting Items for Systematic Reviews and Meta-Analyses (PRISMA) guidelines were followed for the preparation (PROSPERO ID = CRD42021216815, https://www.crd.york.ac.uk/prospero/display_record.php?ID=CRD42021216815) and writing of this review (S1 Table). A comprehensive search strategy for relevant articles was applied in several electronic databases including Pubmed, Embase, Web of Science, and Global Index Medicus. We searched from the date the databases were created to October 2020. The search terms covered exposure (DM) and outcome (TB) (S2 Table). Beyond this electronic search, we performed an additional review of the bibliographic references of relevant works for additional inclusions.

### Inclusion and non-inclusion criteria

We included in the present review, all observational studies (cohort, case-control and cross-sectional) which investigated the association between DM and TB without any restriction by geographic location, time and DM and TB diagnostic approaches. The studies included were those written in English or French. Excluded from this review were studies for which we did not have access to the abstract and/or full-text, duplicates, studies with designs or populations inappropriate for the purposes of the present work.

### Study selection and data extraction

The results of the manual and electronic search were screened by two investigators (JETB and SK) using the Rayyan review application. Eligibility and data extraction from full texts were carried out by all investigators in this review. The following parameters were extracted from the included articles: first author, year of publication, study design, sampling approach, timing (retrospectively/prospectively) of (exposure follow up, timing of DM and TB testing), country, study period and duration, age range of participants, DM and TB testing approaches, DM and TB case definition, inclusion and exclusion criteria, pairing parameters, data on qualitative and quantitative confounding factors and data on the total numbers of cases (diabetic or TB) and controls. Qualitative confounders included gender, smoking, alcohol consumption, HIV infection, malignant diseases, chronic kidney diseases, and several other socio-demographic

and co-morbidities. Quantitative confounders included age, body mass index, and several other blood components. Discussion and consensus among investigators were used if there were any disagreement.

### Quality assessment

The quality of the included observational studies was assessed according to the Joanna Briggs Institute scale (S3 Table) [27]. The cross-sectional, case-control and cohort studies consisted of 8, 10 and 11 questions respectively with the expected answers being (Yes, No, Unclear or Not applicable). We attributed 1 mark for the answers (Yes) and 0 for the other answers (No, Unclear and Not applicable). We rated studies as having low, moderate, and high risk of bias according to total marks per study. All investigators in this study independently collected answers to the Joanna Briggs Institute scale questions in duplicate. Disagreements were resolved by discussion and consensus.

### Statistical analysis

We opted to group the results according to study designs (cross-sectional, control cases, and cohorts). We selected the data from the reference methods (culture for active TB, IGRA for latent TB, and OGTT for DM) in studies reporting multiple data on the relationship between DM and TB for the same population. The odds ratio (OR), its confidence interval (95% CI) and the prediction interval were calculated using random-effects models on the R software version 4.0.3 [28]. Egger's test ($< 0.1$) and funnel charts (asymmetric distribution) were used to indicate the existence of publication bias [29]. The Chi-square test and the I2 and H indices were used to estimate heterogeneity in the estimates [30]. Subgroup analyses and metaregression were used to investigate the parameters responsible for the heterogeneity. Parameters included in these subgroup analyses included sampling method, number of study sites, timing of DM and TB testing, country, country income level [31], and study duration. P values $< 0.05$ indicated statistical significance. Sensitivity analyses that included only studies with a low risk of bias and studies comparable with regard to confounding factors were performed to enhance the accuracy of our results. We determined the comparability of studies with confounding factors using Chi-square, Fisher or Student's T-test as reported previously [32].

## Results

### Study selection

The electronic search yielded 12,742 articles from PubMed (6725), Web of Science (6123), Embase (693), and Global Index Medicus (201). Manual search yielded 54 additional articles (Fig 1). From these, the eligibility review resulted in 201 articles and the exclusion brought this number to 154 (S4 Table) and finally the inclusion resulted 47 articles used (49 effect estimates) in this review [33–79].

### Summary characteristics of included studies

The detailed description of the individual characteristics of the included studies is presented in Table 1. The included studies were published between 1992 and 2020. Cases (TB), controls (non-TB), exposed (diabetics) and unexposed (non-diabetics) were recruited from 1976 to 2018. The majority of studies had a case-control design (23/49), while 16 were cross-sectional and 10 cohort studies. No investigator of the included studies performed a prospective follow-up of exposed/unexposed subjects in the included studies. Five studies were representative of the national population. Included studies were performed in 18 countries spread across

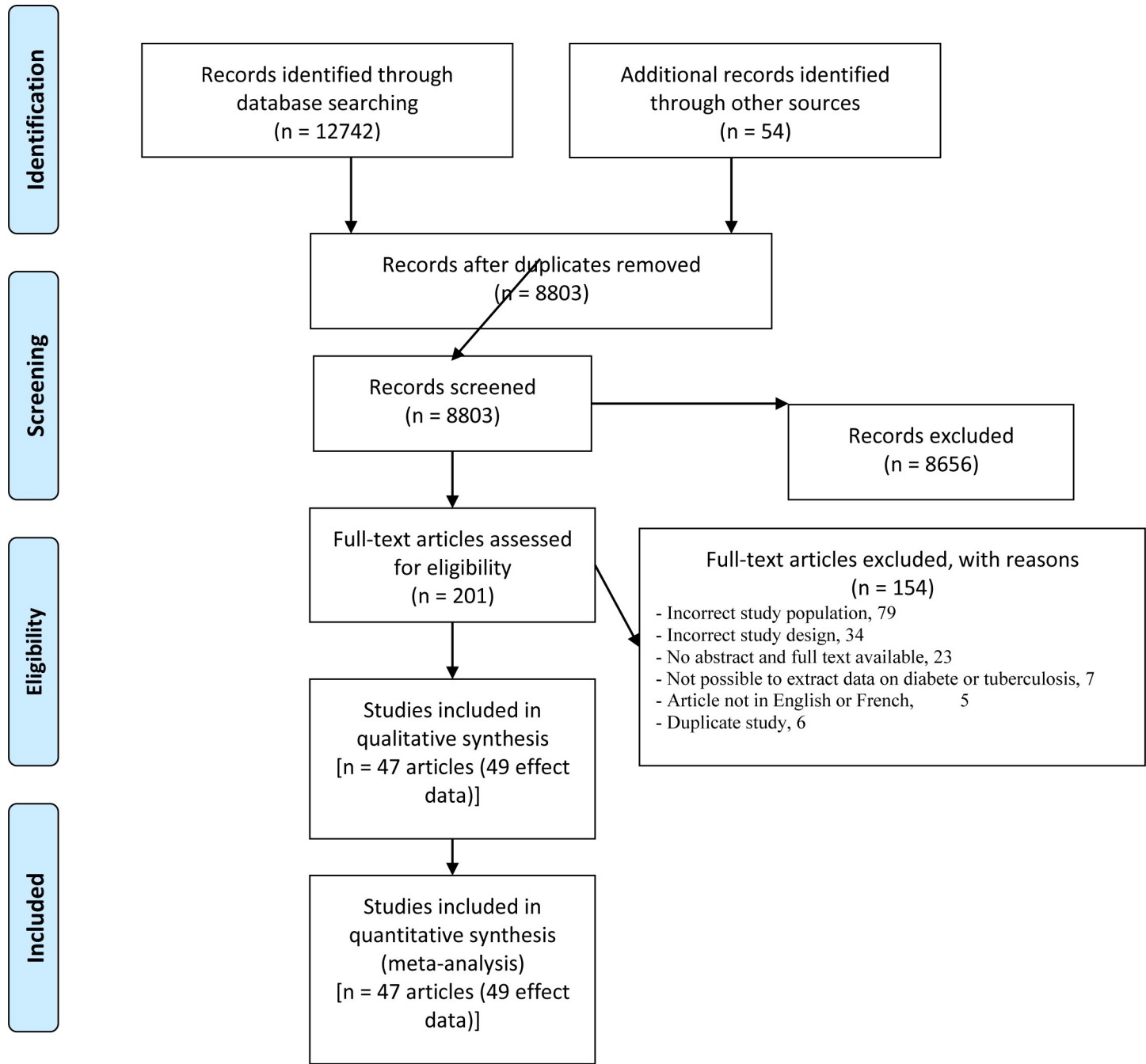

**Fig 1. PRISMA flow-chart of studies selected for the meta-analysis.**

different regions of the world and more particularly in China (16/49) and the United States of America (11/49). High-income countries (26/49) were the most represented and only one study was conducted in low-income countries [46]. The vast majority of studies recruited adults over 15 years old. Apart from studies with multiple diagnostic methods, the International Classification of Diseases (ICD) code was the most widely used approach for DM (10/26) and TB (11/20). Nineteen included studies paired reference with controls with at least one parameter.

**Table 1. Individual characteristics of included studies.**

| Study Design | Country | Study period | TB stage | TB diagnosis approach | DM diagnosis approach | Controls | Matched parameters between cases and controls | Author, Year |
|---|---|---|---|---|---|---|---|---|
| Case control | Indonesia | Mar/2001-Mar/2005 | Active TB | Clinical, chest X-rays, Microscopy | Fasting blood glucose | Presumed healthy controls | Age, Gender, county of residence | Alisjahbana, 2006 [33] |
| Cross sectional | China | Aug/2001-Dec/2004 | Active TB | ICD code, Medical records | ICD code, Medical records | Non-DM | Unclear/ Not reported | Baker, 2012 [34] |
| Cross sectional | United States of America | 2011–2012 | Latent TB infection | IGRA Test, Tuberculin skin test | Doctor-diagnosed DM, Glycated hemoglobin A1c test | Non-TB diseases | Unclear/ Not reported | Barron, 2018 –DM [35] |
| Cross sectional | United States of America | 2011–2012 | Latent TB infection | IGRA Test, Tuberculin skin test | Doctor-diagnosed DM, Glycated hemoglobin A1c test | Non-TB diseases | Unclear/ Not reported | Barron, 2018 – PreDM [35] |
| Case control | Tanzania | Jun/2012-Dec/2013 | Active TB | Clinical, chest X-rays, Microscopy | Fasting blood glucose, Oral glucose tolerance test, Glycated hemoglobin A1c test | Presumed healthy controls | Age, Gender | Boillat-blanco, 2016 [36] |
| Case control | United States of America | Sep/1998-Dec/2003 | Active TB | ICD code | ICD code | Presumed healthy controls | Unclear/ Not reported | Brassard, 2006 [37] |
| Case control | United States of America | 1988–1990 | Active TB | Microscopy, Culture, PCR | Unclear/ Not reported | Non-TB diseases | Unclear/ Not reported | Buskin, 1994 [38] |
| Cross sectional | China | Jan/1983- Dec/2003 | Active TB | Clinical, chest X-rays, Culture | Unclear/ Not reported | Non-TB diseases | Unclear/ Not reported | Chen, 2006 [39] |
| Case control | China | 1997–2010 | Active TB | ICD code | ICD code | Presumed healthy controls | Age, Gender, Recruitment time | Chung, 2014 [40] |
| Case control | United States of America | 1976–1980 | Active TB | Doctor-diagnosed TB | Doctor-diagnosed DM, Oral glucose tolerance test | Non-TB diseases | Unclear/ Not reported | Corris, 2012 [41] |
| Case control | Kazakhstan | Jun/ 2012-May/ 2014 | Active TB | Clinical, chest X-rays, Culture | Doctor-diagnosed DM | Presumed healthy controls | County of residence | Davis, 2017 [42] |
| Case control | Tanzania | Apr/2006-Jan/2009 | Active TB | Microscopy, Culture | Fasting blood glucose, Oral glucose tolerance test | Presumed healthy controls | Age, Gender | Faurholt-Jepsen, 2011 [44] |
| Cross sectional | Tanzania | Apr/ 2006—Mar/ 2009 | Active TB | Culture | Fasting blood glucose, Oral glucose tolerance test | Presumed healthy controls | Age, Gender | Faurholt-Jepsen, 2014 [43] |
| Cohort | United States of America | Jan/ 2001—Dec/ 2011 | Active TB | chest X-rays | ICD code, Fasting blood glucose | Non-DM | Unclear/ Not reported | Golub, 2019 [45] |
| Case control | Guinea-Bissau | July/2010-July/2011 | Active TB | Clinical, chest X-rays, Microscopy | Fasting blood glucose, Random blood sugar test | Presumed healthy controls | Unclear/ Not reported | Haraldsdottir, 2015 [46] |
| Cross sectional | United States of America | October/2013-August/2014 | Latent TB infection | chest X-rays, IGRA Test | Glycated hemoglobin A1c test | Non-TB diseases | Unclear/ Not reported | Hensel, 2016 [47] |
| Case control | Bangladesh | Jan/2008-Jul/2008 | Active TB | Microscopy | Oral glucose tolerance test | Non-TB diseases | Unclear/ Not reported | Hossain, 2014 [48] |
| Case control | United Kingdom | 1990–2001 | Active TB | Medical records | Medical records | Presumed healthy controls | Age, Gender, County of residence | Jick, 2006 [49] |
| Case control | Croatia | 2006–2008 | Active TB | Culture | Unclear/ Not reported | Non-TB diseases | Age, Gender, county of residence | Jurcev-Savicevic, 2013 [50] |

(*Continued*)

**Table 1.** (Continued)

| Study Design | Country | Study period | TB stage | TB diagnosis approach | DM diagnosis approach | Controls | Matched parameters between cases and controls | Author, Year |
|---|---|---|---|---|---|---|---|---|
| Cross sectional | Thailand | Mar/2012-Mar/2013 | Latent TB infection | Tuberculin skin test, IGRA Test | Unclear/ Not reported | Presumed healthy controls | Unclear/ Not reported | Khawcharoenporn, 2015 [51] |
| Cohort | Korean | 1988–1990 | Active TB | chest X-rays, Microscopy, Culture | Glucose oxidase test | Non-DM | Unclear/ Not reported | Kim, 1995 [52] |
| Cross sectional | India | May/2014-Nov/2015 | Active TB | Tuberculin skin test, Microscopy, Culture | Clinical, Random blood sugar test | Presumed healthy controls | Unclear/ Not reported | Kubiak, 2019 [53] |
| Cohort | China | 2000–2011 | Active TB | ICD code | ICD code | Presumed healthy controls | Age, Gender | Kuo, 2013 [54] |
| Case control | China | 1998–2011 | Active TB | ICD code | ICD code | Non-TB diseases | Age, Gender | Lai, 2014 [55] |
| Cohort | China | 1997–2007 | Active TB | ICD code | ICD code | Non-DM | Age, Gender, Recruitment time | Lee, 2013 [56] |
| Case control | China | 2006–2017 | Active TB | Clinical, Medical records, chest X-rays, Microscopy, Culture | ICD code, Medical records, Fasting blood glucose, Glycated hemoglobin A1c test | Non-TB diseases | Unclear/ Not reported | Lee, 2014 [58] |
| Cohort | China | Mar/2005-Dec/2012 | Active TB | ICD code, Medical records | Fasting blood glucose | Non-DM | Unclear/ Not reported | Lee, 2016 [57] |
| Case control | Denmark | Jan/1980-Dec/2008 | Active TB | ICD code | Clinical, Medical records, Glycated hemoglobin A1c test | Non-TB diseases | Age, Gender, county of residence | Leegaard, 2011 [59] |
| Cohort | China | Jan/2000—Dec/2005 | Active TB | Clinical, Medical records, chest X-rays, Histopathology, Culture | Glycated hemoglobin A1c test | Non-DM | Unclear/ Not reported | Leung, 2008 [60] |
| Cohort | China | 2000–2009 | Active TB | ICD code | ICD code | Non-DM | Age, Gender, Recruitment time | Lin, 2017 [62] |
| Cohort | China | 2005–2013 | Latent TB infection | Clinical, chest X-rays, Tuberculin skin test, IGRA Test | Doctor-diagnosed DM | Non-DM | Unclear/ Not reported | Lin, 2019 [61] |
| Case control | India | Jan/1983-Dec/1989 | Active TB | Tuberculin skin test | Medical records, Fasting blood glucose, Any glucose level | Non-TB diseases | Unclear/ Not reported | Mori, 1992 [63] |
| Case control | Romania | Mar/2014—Mar/2015 | Active TB | Microscopy, Culture, PCR | Unclear/ Not reported | Non-TB diseases | Age, Gender, county of residence | Ndishimye, 2017 [64] |
| Case control | United States of America | 1991 | Active TB | ICD code | ICD code | Non-TB diseases | Unclear/ Not reported | Pablos-Méndez, 1997 [65] |
| Cohort | United Kingdom | Jan/1990-Dec/2012 | Active TB | ICD code | ICD code | Non-DM | Age, Gender | Pealing, 2015 [66] |
| Case control | Brazil | Aug/2008-Apr/2010 | Active TB | Clinical, Microscopy, Culture | Fasting blood glucose, Oral glucose tolerance test | Non-TB diseases | Age, Gender | Pereira, 2016 [67] |
| Case control | United States of America | 1999–2001 | Active TB | ICD code | ICD code | Non-TB diseases | Unclear/ Not reported | Pérez, 2006 [68] |
| Cohort | China | 2002–2011 | Active TB | ICD code | ICD code | Non-DM | Gender | Shen, 2014 [69] |
| Case control | Japan | Jan/2015-Dec/2018 | Active TB | Clinical, chest X-rays, IGRA Test, Microscopy, Culture, PCR | Doctor-diagnosed DM | Non-TB diseases | County of residence | Shimouchi, 2020 [70] |

(*Continued*)

**Table 1.** (Continued)

| Study Design | Country | Study period | TB stage | TB diagnosis approach | DM diagnosis approach | Controls | Matched parameters between cases and controls | Author, Year |
|---|---|---|---|---|---|---|---|---|
| Cross sectional | China | Mar/2011-Feb/2012 | Latent TB infection | ELISA, Microscopy, Culture | Unclear/ Not reported | Non-TB diseases | Unclear/ Not reported | Shu, 2012 [71] |
| Cross sectional | United States of America | Apr/2005-Mar/2012 | Latent TB infection | Tuberculin skin test, IGRA Test | Medical records | Non-DM | Unclear/ Not reported | Suwanpimolkul, 2014 [72] |
| Cross sectional | United States of America | Apr/2005-Mar/2012 | Active TB | Tuberculin skin test, IGRA Test | Medical records | Non-DM | Unclear/ Not reported | Suwanpimolkul, 2014 [72] |
| Cross sectional | Malaysia | Oct/2014-Dec/2015 | Latent TB infection | Clinical, chest X-rays, Tuberculin skin test, Microscopy | Fasting blood glucose, Glycated hemoglobin A1c test, Random blood sugar test | Non-DM | Unclear/ Not reported | Swarna Nantha, 2017 [73] |
| Cross sectional | China | Jan/2011-Dec/2012 | Latent TB infection | Medical records, chest X-rays, IGRA Test | Unclear/ Not reported | Non-TB diseases | Unclear/ Not reported | Ting, 2014 [74] |
| Case control | Republic of Kiribati | Jun/2010-Mar/2012 | Latent TB infection | Clinical, Doctor-diagnosed DM, chest X-rays, Tuberculin skin test, Microscopy, Culture | Glycated hemoglobin A1c test | Non-TB diseases | Unclear/ Not reported | Viney, 2015 [75] |
| Cross sectional | China | Sep/2010-Dec/2012 | Active TB | Clinical, chest X-rays, Microscopy | Fasting blood glucose | Non-TB diseases | County of residence | Wang, 2013 [76] |
| Cross sectional | Indonesia | 2014–2015 | Active TB | Doctor-diagnosed DM | Doctor-diagnosed DM | Non-TB diseases | Unclear/ Not reported | Wardhani, 2019 [77] |
| Cross sectional | China | Jan/2002-Dec/2004 | Active TB | Culture | Medical records | Presumed healthy controls | Unclear/ Not reported | Wu, 2007 [78] |
| Case control | Kazakhstan | Jun/2012-Jan/2013 | Active TB | Clinical, chest X-rays, Microscopy, Culture, PCR | Unclear/ Not reported | Presumed healthy controls | Age | Zhussupov, 2016 [79] |

DM: Diabetes Mellitus; ICD: International Classification of Diseases; TB: Tuberculosis.

## Risk of bias in included studies

The methodological quality of the included studies is shown in S5 Table. Overall, the included studies had a low risk of bias (32/49). Most of the included studies collected data and considered confounding factors in the analysis of the association between DM and the TB development risk. In cohort studies, diabetic and nondiabetic patients were generally recruited from the same population, diagnosed with DM and TB in the same way, tested for absence of TB at the start of the follow-up, and followed up with a completeness rate. TB and non-TB patients recruited from case control studies were generally comparable and diagnosed with TB and DM in the same way. In cross-sectional studies, the study context and inclusion criteria for participants were well defined.

## Meta-analysis

In this meta-analysis, 503,760 cases (diabetic or TB) and 3,596,845 controls were considered to calculate the combined effect of the association between DM and the TB risk. Regarding the study design, the 49-effect estimate showed an increased risk of developing TB in diabetic patients (OR = 2.3, 95% CI = [2.0–2.7]) (Fig 2). This overall effect was associated with substantial heterogeneity (I2 = 94.2% [93.0–95.1]). The association between DM and the risk of developing TB was conserved in the 10 cohort (OR = 2.0, CI 95% = [1.5–2.4]), the 23 case-control

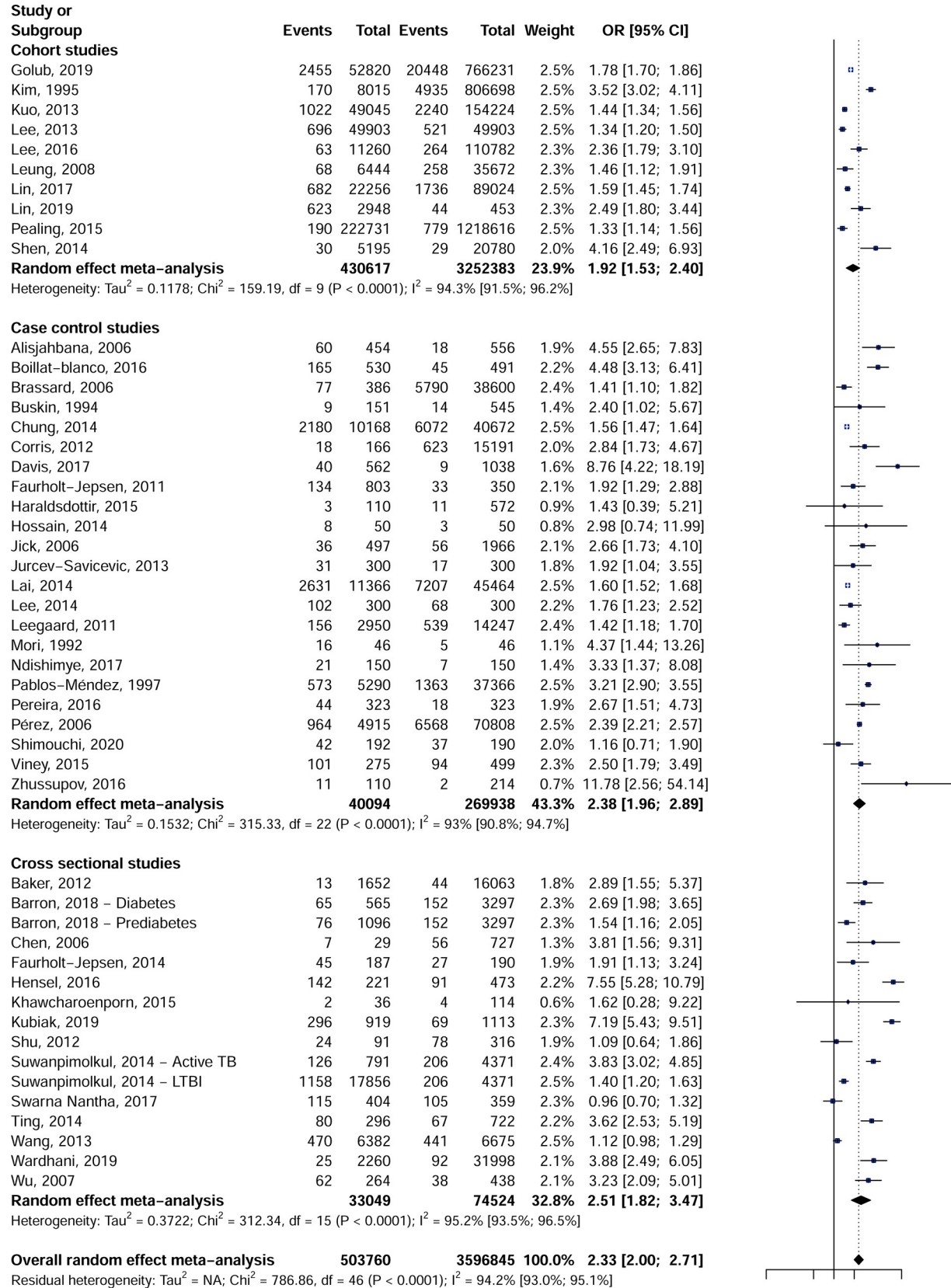

| Study or Subgroup | Events | Total | Events | Total | Weight | OR [95% CI] |
|---|---|---|---|---|---|---|
| **Cohort studies** | | | | | | |
| Golub, 2019 | 2455 | 52820 | 20448 | 766231 | 2.5% | 1.78 [1.70; 1.86] |
| Kim, 1995 | 170 | 8015 | 4935 | 806698 | 2.5% | 3.52 [3.02; 4.11] |
| Kuo, 2013 | 1022 | 49045 | 2240 | 154224 | 2.5% | 1.44 [1.34; 1.56] |
| Lee, 2013 | 696 | 49903 | 521 | 49903 | 2.5% | 1.34 [1.20; 1.50] |
| Lee, 2016 | 63 | 11260 | 264 | 110782 | 2.3% | 2.36 [1.79; 3.10] |
| Leung, 2008 | 68 | 6444 | 258 | 35672 | 2.3% | 1.46 [1.12; 1.91] |
| Lin, 2017 | 682 | 22256 | 1736 | 89024 | 2.5% | 1.59 [1.45; 1.74] |
| Lin, 2019 | 623 | 2948 | 44 | 453 | 2.3% | 2.49 [1.80; 3.44] |
| Pealing, 2015 | 190 | 222731 | 779 | 1218616 | 2.5% | 1.33 [1.14; 1.56] |
| Shen, 2014 | 30 | 5195 | 29 | 20780 | 2.0% | 4.16 [2.49; 6.93] |
| **Random effect meta-analysis** | | **430617** | | **3252383** | **23.9%** | **1.92 [1.53; 2.40]** |

Heterogeneity: $Tau^2 = 0.1178$; $Chi^2 = 159.19$, df = 9 (P < 0.0001); $I^2 = 94.3\%$ [91.5%; 96.2%]

| Study or Subgroup | Events | Total | Events | Total | Weight | OR [95% CI] |
|---|---|---|---|---|---|---|
| **Case control studies** | | | | | | |
| Alisjahbana, 2006 | 60 | 454 | 18 | 556 | 1.9% | 4.55 [2.65; 7.83] |
| Boillat-blanco, 2016 | 165 | 530 | 45 | 491 | 2.2% | 4.48 [3.13; 6.41] |
| Brassard, 2006 | 77 | 386 | 5790 | 38600 | 2.4% | 1.41 [1.10; 1.82] |
| Buskin, 1994 | 9 | 151 | 14 | 545 | 1.4% | 2.40 [1.02; 5.67] |
| Chung, 2014 | 2180 | 10168 | 6072 | 40672 | 2.5% | 1.56 [1.47; 1.64] |
| Corris, 2012 | 18 | 166 | 623 | 15191 | 2.0% | 2.84 [1.73; 4.67] |
| Davis, 2017 | 40 | 562 | 9 | 1038 | 1.6% | 8.76 [4.22; 18.19] |
| Faurholt-Jepsen, 2011 | 134 | 803 | 33 | 350 | 2.1% | 1.92 [1.29; 2.88] |
| Haraldsdottir, 2015 | 3 | 110 | 11 | 572 | 0.9% | 1.43 [0.39; 5.21] |
| Hossain, 2014 | 8 | 50 | 3 | 50 | 0.8% | 2.98 [0.74; 11.99] |
| Jick, 2006 | 36 | 497 | 56 | 1966 | 2.1% | 2.66 [1.73; 4.10] |
| Jurcev-Savicevic, 2013 | 31 | 300 | 17 | 300 | 1.8% | 1.92 [1.04; 3.55] |
| Lai, 2014 | 2631 | 11366 | 7207 | 45464 | 2.5% | 1.60 [1.52; 1.68] |
| Lee, 2014 | 102 | 300 | 68 | 300 | 2.2% | 1.76 [1.23; 2.52] |
| Leegaard, 2011 | 156 | 2950 | 539 | 14247 | 2.4% | 1.42 [1.18; 1.70] |
| Mori, 1992 | 16 | 46 | 5 | 46 | 1.1% | 4.37 [1.44; 13.26] |
| Ndishimye, 2017 | 21 | 150 | 7 | 150 | 1.4% | 3.33 [1.37; 8.08] |
| Pablos-Méndez, 1997 | 573 | 5290 | 1363 | 37366 | 2.5% | 3.21 [2.90; 3.55] |
| Pereira, 2016 | 44 | 323 | 18 | 323 | 1.9% | 2.67 [1.51; 4.73] |
| Pérez, 2006 | 964 | 4915 | 6568 | 70808 | 2.5% | 2.39 [2.21; 2.57] |
| Shimouchi, 2020 | 42 | 192 | 37 | 190 | 2.0% | 1.16 [0.71; 1.90] |
| Viney, 2015 | 101 | 275 | 94 | 499 | 2.3% | 2.50 [1.79; 3.49] |
| Zhussupov, 2016 | 11 | 110 | 2 | 214 | 0.7% | 11.78 [2.56; 54.14] |
| **Random effect meta-analysis** | | **40094** | | **269938** | **43.3%** | **2.38 [1.96; 2.89]** |

Heterogeneity: $Tau^2 = 0.1532$; $Chi^2 = 315.33$, df = 22 (P < 0.0001); $I^2 = 93\%$ [90.8%; 94.7%]

| Study or Subgroup | Events | Total | Events | Total | Weight | OR [95% CI] |
|---|---|---|---|---|---|---|
| **Cross sectional studies** | | | | | | |
| Baker, 2012 | 13 | 1652 | 44 | 16063 | 1.8% | 2.89 [1.55; 5.37] |
| Barron, 2018 – Diabetes | 65 | 565 | 152 | 3297 | 2.3% | 2.69 [1.98; 3.65] |
| Barron, 2018 – Prediabetes | 76 | 1096 | 152 | 3297 | 2.3% | 1.54 [1.16; 2.05] |
| Chen, 2006 | 7 | 29 | 56 | 727 | 1.3% | 3.81 [1.56; 9.31] |
| Faurholt-Jepsen, 2014 | 45 | 187 | 27 | 190 | 1.9% | 1.91 [1.13; 3.24] |
| Hensel, 2016 | 142 | 221 | 91 | 473 | 2.2% | 7.55 [5.28; 10.79] |
| Khawcharoenporn, 2015 | 2 | 36 | 4 | 114 | 0.6% | 1.62 [0.28; 9.22] |
| Kubiak, 2019 | 296 | 919 | 69 | 1113 | 2.3% | 7.19 [5.43; 9.51] |
| Shu, 2012 | 24 | 91 | 78 | 316 | 1.9% | 1.09 [0.64; 1.86] |
| Suwanpimolkul, 2014 – Active TB | 126 | 791 | 206 | 4371 | 2.4% | 3.83 [3.02; 4.85] |
| Suwanpimolkul, 2014 – LTBI | 1158 | 17856 | 206 | 4371 | 2.5% | 1.40 [1.20; 1.63] |
| Swarna Nantha, 2017 | 115 | 404 | 105 | 359 | 2.3% | 0.96 [0.70; 1.32] |
| Ting, 2014 | 80 | 296 | 67 | 722 | 2.2% | 3.62 [2.53; 5.19] |
| Wang, 2013 | 470 | 6382 | 441 | 6675 | 2.5% | 1.12 [0.98; 1.29] |
| Wardhani, 2019 | 25 | 2260 | 92 | 31998 | 2.1% | 3.88 [2.49; 6.05] |
| Wu, 2007 | 62 | 264 | 38 | 438 | 2.1% | 3.23 [2.09; 5.01] |
| **Random effect meta-analysis** | | **33049** | | **74524** | **32.8%** | **2.51 [1.82; 3.47]** |

Heterogeneity: $Tau^2 = 0.3722$; $Chi^2 = 312.34$, df = 15 (P < 0.0001); $I^2 = 95.2\%$ [93.5%; 96.5%]

| **Overall random effect meta-analysis** | | **503760** | | **3596845** | **100.0%** | **2.33 [2.00; 2.71]** |

Residual heterogeneity: $Tau^2 = NA$; $Chi^2 = 786.86$, df = 46 (P < 0.0001); $I^2 = 94.2\%$ [93.0%; 95.1%]

**Fig 2. Association between diabetes mellitus and risk of tuberculosis in cohort, case control and cross-sectional studies.**

(OR = 2.4, CI 95% = [2.0–2.9]) and 16 cross-sectional studies (OR = 2.5, CI 95% = [1.8–3.5]). A significant publication bias was recorded in the cross-sectional (p Egger = 0.058) and the case-control studies (p Egger = 0.093) unlike the cohort studies (p Egger = 0.417) which did not present any publication bias (Table 2, S1–S3 Figs). Considering only studies with low risk of bias sensitivity analysis did not reveal any difference from the overall results. The data collected for 81 qualitative variables and 16 quantitative variables considered to be confounding factors enabled us to select studies that had similar proportions in references and controls (S6 and S7 Tables). For cohort studies, sensitivity analyses including only comparable studies for confounding factors showed similar results to overall results, including factors such as HIV infection, malignancies and age. For the case-control studies, the same trend was observed for the sensitivity analysis including only comparable studies mainly for alcohol drinkers, chronic kidney disease, drug users, HIV infected patients, tobacco exposure, and age. For the cross-sectional studies, on the other hand, the overall effect observed was lost for the sensitivity analyses including only comparable studies for certain confounding factors including chronic kidney disease, patients with cirrhosis of the liver or malignant diseases.

## Source of heterogeneity examination

The potential sources of heterogeneity were explored by the subgroup analyses. These sources included country, UNSD region, country income level, TB stage (active vs latent), and type of controls (S8 Table). In the cohort, control and cross-sectional designs only the geographic location (countries and UNSD regions) contributed to a source of heterogeneity (p subgroup difference <0.05). In cohort studies, however, all subcategories showed an association between DM and the risk of developing TB.

## Discussion

This systematic review included 47 articles examining the association between DM and TB. Regardless of study design, region of origin, stage of TB (latent or active TB), type of controls (non-DM, non-TB, or presumed healthy), this meta-analysis suggests that DM increases the risk of developing TB. The overall effect observed suggests that patients with DM are two times more likely to develop TB than non-diabetics. This overall effect persisted in the sensitivity analysis including only studies with similar proportions of common confounders between cases and controls.

The statistically significant association between DM and TB observed in this review is consistent with those reported previously. A first qualitative review in 2007 with 9 included studies reported effect estimates ranging from 1.5 to 7.8 fold the risk of TB in DM patients [80]. Two other meta-analyses that included studies with patients with active TB and whose age-adjusted estimates were reported in 2008 and 2018 [14, 24]. One of these meta-analyses reported an estimated 3.1-fold effect for 3 cohort studies and the second an estimated 1.5-fold effect for 14 studies with low risk of bias. A final meta-analysis with studies recruiting patients with latent TB revealed no significant association for one cohort study and a weak association for 12 cross-sectional studies [16]. Compared to these previous systematic reviews, we included over 10 additional articles and used a very rigorous methodology including calculating effect estimates of primary data from included studies and taking into account a wide range of confounding factors of the association between DM and TB listed in the articles included [35, 43, 45, 48, 53, 58, 61, 62, 64, 67, 70, 73, 77, 79]. Little is known about the biological mechanisms that underlie a high risk of developing TB in patients with DM. Several hypotheses linked to an alteration of immune function in diabetics have however been suggested to explain this association between DM and TB [81–84]. These hypotheses include, but not limited to:

**Table 2. TB development in people with and without DM and influence of confounders.**

| | OR (95% CI) | 95% Prediction interval | N Studies | N LRTI cases | N controls | H (95% CI) | I² (95%CI) | P heterogeneity | P Egger test |
|---|---|---|---|---|---|---|---|---|---|
| **Cohort studies** | | | | | | | | | |
| Overall | 1.9 [1.5–2.4] | [0.8–4.4] | 10 | 430617 | 3252383 | 4.2 [3.4–5.2] | 94.3 [91.5–96.2] | < 0.001 | 0.417 |
| Low risk of bias | 1.6 [1.4–1.7] | [1.1–2.2] | 7 | 414459 | 2424452 | 2.9 [2.1–4] | 88.2 [78.2–93.7] | < 0.001 | 0.496 |
| Asbestosis | 1.6 [1.5–1.7] | NA | 1 | 22256 | 89024 | NA | NA | 1 | NA |
| Autoimmune disorders | 1.3 [1.2–1.5] | NA | 1 | 49903 | 49903 | NA | NA | 1 | NA |
| Bet nut use | 2.4 [1.8–3.1] | NA | 1 | 11260 | 110782 | NA | NA | 1 | NA |
| Chronic kidney disease | 1.8 [1.7–1.9] | NA | 1 | 52820 | 766231 | NA | NA | 1 | NA |
| HIV infection | 1.5 [1.3–1.7] | NA | 2 | 72159 | 138927 | 2.3 [1.1–4.7] | 81 [19.2–95.6] | 0.022 | NA |
| Male gender | 1.8 [1.2–2.5] | [0.3–9.4] | 4 | 83798 | 195379 | 2.6 [1.7–4.1] | 85.4 [63.9–94.1] | < 0.001 | 0.377 |
| Malignant disease | 1.8 [1.7–1.8] | NA | 2 | 59264 | 801903 | 1.4 | 49 | 0.161 | NA |
| Pneumoconiosis | 1.6 [1.5–1.7] | NA | 1 | 22256 | 89024 | NA | NA | 1 | NA |
| Age | 1.5 [1.3–1.7] | NA | 2 | 72159 | 138927 | 2.3 [1.1–4.7] | 81 [19.2–95.6] | 0.022 | NA |
| **Case control studies** | | | | | | | | | |
| Overall | 2.4 [2–2.9] | [1–5.5] | 23 | 40094 | 269938 | 3.8 [3.3–4.4] | 93 [90.8–94.7] | < 0.001 | 0.093 |
| Low risk of bias | 2.2 [1.7–2.9] | [0.8–6.2] | 13 | 28831 | 144497 | 2.6 [2.1–3.3] | 85.4 [76.6–90.9] | < 0.001 | 0.05 |
| Adenotonsillectomy | 1.6 [1.5–1.7] | NA | 1 | 11366 | 45464 | NA | NA | 1 | NA |
| Central sewage system | 1.9 [1–3.5] | NA | 1 | 300 | 300 | NA | NA | 1 | NA |
| Chronic kidney disease | 1.9 [1.4–2.6] | [0.3–14] | 3 | 710 | 814 | 1.7 [1–3.1] | 64.7 [0–89.9] | 0.059 | 0.271 |
| Co_morbidity | 1.9 [1–3.5] | NA | 1 | 300 | 300 | NA | NA | 1 | NA |
| Currently rent home | 11.8 [2.6–54.1] | NA | 1 | 110 | 214 | NA | NA | 1 | NA |
| Drinker | 3.2 [2.9–3.5] | [2.5–3.9] | 4 | 5850 | 38388 | 1.2 [1–1.9] | 26.3 [0–72.1] | 0.254 | 0.101 |
| Drug user | 3.8 [1.8–7.9] | [0–16592.3] | 3 | 1012 | 1488 | 2.2 [1.2–3.9] | 79.5 [34.7–93.5] | 0.008 | 0.916 |
| Ever injected heroin | 8.8 [4.2–18.2] | NA | 1 | 562 | 1038 | NA | NA | 1 | NA |
| Ever smoked heroin | 8.8 [4.2–18.2] | NA | 1 | 562 | 1038 | NA | NA | 1 | NA |
| Ever used opium | 8.8 [4.2–18.2] | NA | 1 | 562 | 1038 | NA | NA | 1 | NA |
| Extra pulmonary lesion | 1.8 [1.2–2.5] | NA | 1 | 300 | 300 | NA | NA | 1 | NA |
| Family history of diabetes mellitus | 3 [0.7–12] | NA | 1 | 50 | 50 | NA | NA | 1 | NA |
| Hepatitis C infection, Anti_HCV | 11.8 [2.6–54.1] | NA | 1 | 110 | 214 | NA | NA | 1 | NA |

*(Continued)*

**Table 2.** (Continued)

| | OR (95% CI) | 95% Prediction interval | N Studies | N LRTI cases | N controls | H (95% CI) | I² (95%CI) | P heterogeneity | P Egger test |
|---|---|---|---|---|---|---|---|---|---|
| HIV infection | 1.5 [1.3–1.8] | [0.5–4.6] | 3 | 3360 | 14761 | 2 [1.1–3.6] | 74.9 [16.8–92.4] | 0.019 | 0.277 |
| Hyperlipidaemia | 1.6 [1.5–1.6] | NA | 1 | 10168 | 40672 | NA | NA | 1 | NA |
| Illicit drug use | 1.9 [1–3.5] | NA | 1 | 300 | 300 | NA | NA | 1 | NA |
| Immunosuppressive therapy | 1.9 [1–3.5] | NA | 1 | 300 | 300 | NA | NA | 1 | NA |
| Living in a crowded home | 4.6 [2.6–7.8] | NA | 1 | 454 | 556 | NA | NA | 1 | NA |
| Male gender | 2.1 [1.6–2.7] | [0.9–4.9] | 10 | 26090 | 117338 | 2.1 [1.5–2.8] | 77.2 [58.2–87.6] | < 0.001 | 0.036 |
| Malignant disease | 1.8 [1.2–2.5] | NA | 1 | 300 | 300 | NA | NA | 1 | NA |
| Marital status, Single | 2.1 [1.5–3.1] | NA | 2 | 953 | 500 | 1.1 | 17.3 | 0.272 | NA |
| Other chronic diseases | 1.9 [1–3.5] | NA | 1 | 300 | 300 | NA | NA | 1 | NA |
| Pancreatitis | 2.4 [1–5.7] | NA | 1 | 151 | 545 | NA | NA | 1 | NA |
| Physical activity | 3 [0.7–12] | NA | 1 | 50 | 50 | NA | NA | 1 | NA |
| Poly_drug resistant | 1.8 [1.2–2.5] | NA | 1 | 300 | 300 | NA | NA | 1 | NA |
| Previous hospitalizations | 1.9 [1–3.5] | NA | 1 | 300 | 300 | NA | NA | 1 | NA |
| Prisoners | 1.9 [1–3.5] | NA | 1 | 300 | 300 | NA | NA | 1 | NA |
| Smoke Exposure | 2.5 [2–3.3] | [1.4–4.4] | 4 | 702 | 16807 | 1 [1–1.5] | 0 [0–53.1] | 0.806 | 0.86 |
| Transplantation | 1.9 [1–3.5] | NA | 1 | 300 | 300 | NA | NA | 1 | NA |
| Age | 2.4 [1.6–3.7] | [0.6–10.1] | 5 | 4756 | 15561 | 2.9 [2–4.3] | 88.5 [75.8–94.5] | < 0.001 | 0.393 |
| Cigarettes smoked in a week | 8.8 [4.2–18.2] | NA | 1 | 562 | 1038 | NA | NA | 1 | NA |
| **Cross sectional studies** | | | | | | | | | |
| Overall | 2.5 [1.8–3.5] | [0.6–9.7] | 16 | 33049 | 74524 | 4.6 [3.9–5.3] | 95.2 [93.5–96.5] | < 0.001 | 0.058 |
| Low risk of bias | 2.4 [1.6–3.5] | [0.5–11.4] | 12 | 30309 | 41171 | 5.2 [4.4–6.1] | 96.3 [94.8–97.3] | < 0.001 | 0.15 |
| Anemia | 1.1 [0.6–1.9] | NA | 1 | 91 | 316 | NA | NA | 1 | NA |
| Atrial fibrillation | 1 [0.7–1.3] | NA | 1 | 404 | 359 | NA | NA | 1 | NA |
| Autoimmune disorders | 2.1 [0.9–4.7] | NA | 2 | 387 | 1038 | 3.7 [2–6.7] | 92.5 [74.7–97.8] | < 0.001 | NA |
| Bronchial asthma | 1 [0.7–1.3] | NA | 1 | 404 | 359 | NA | NA | 1 | NA |
| Bronchiectasis | 3.2 [2.1–5] | NA | 1 | 264 | 438 | NA | NA | 1 | NA |
| Chronic kidney disease | 1.9 [0.7–4.6] | NA | 2 | 700 | 1081 | 5.4 [3.3–8.9] | 96.6 [90.9–98.7] | < 0.001 | NA |
| Chronic liver disease | 1 [0.7–1.3] | NA | 1 | 404 | 359 | NA | NA | 1 | NA |
| Chronic obstructive pulmonary disease | 1 [0.7–1.3] | NA | 1 | 404 | 359 | NA | NA | 1 | NA |
| Drinker | 5 [2.6–9.6] | NA | 2 | 1873 | 16536 | 2.6 [1.3–5.3] | 85.5 [41.6–96.4] | 0.009 | NA |
| Gout | 1 [0.7–1.3] | NA | 1 | 404 | 359 | NA | NA | 1 | NA |
| Health care worker | 3.6 [2.5–5.2] | NA | 1 | 296 | 722 | NA | NA | 1 | NA |
| Hemodialysis patients | 1.9 [0.9–4.1] | NA | 2 | 355 | 754 | 3.1 [1.6–5.9] | 89.5 [60.9–97.2] | 0.002 | NA |

(*Continued*)

**Table 2.** (Continued)

| | OR (95% CI) | 95% Prediction interval | N Studies | N LRTI cases | N controls | H (95% CI) | I² (95%CI) | P heterogeneity | P Egger test |
|---|---|---|---|---|---|---|---|---|---|
| Hepatitis B infection, HBsAg | 2.5 [1.3–4.8] | [0.2–29.5] | 5 | 2315 | 8153 | 4.5 [3.4–6] | 95.1 [91.2–97.3] | < 0.001 | 0.597 |
| Hepatitis C infection, Anti_HCV | 3.5 [1.5–7.8] | [0–71340.3] | 3 | 1346 | 4497 | 4.8 [3.3–7.1] | 95.7 [90.7–98] | < 0.001 | 0.811 |
| HIV infection | 5.2 [3.1–8.7] | NA | 2 | 517 | 1195 | 2.8 [1.4–5.6] | 87.6 [51.8–96.8] | 0.005 | NA |
| Ischaemic heart disease | 1 [0.7–1.3] | NA | 1 | 404 | 359 | NA | NA | 1 | NA |
| Liver cirrhosis | 2.1 [0.9–4.7] | NA | 2 | 387 | 1038 | 3.7 [2–6.7] | 92.5 [74.7–97.8] | < 0.001 | NA |
| Living in a crowded home | 2.9 [1.6–5.4] | NA | 1 | 1652 | 16063 | NA | NA | 1 | NA |
| Male gender | 2.5 [1.6–4] | [0.5–11.9] | 7 | 3841 | 24363 | 3.1 [2.3–4.2] | 89.9 [81.7–94.4] | < 0.001 | 0.883 |
| Malignant disease | 2.6 [1.6–4.3] | [0.3–22.7] | 4 | 680 | 2203 | 2.2 [1.4–3.6] | 79.8 [46.4–92.4] | 0.002 | 0.962 |
| Osteoarthritis | 1 [0.7–1.3] | NA | 1 | 404 | 359 | NA | NA | 1 | NA |
| Residence in an indigenous community | 2.9 [1.6–5.4] | NA | 1 | 1652 | 16063 | NA | NA | 1 | NA |
| Self_reported history of renal failure | 7.2 [5.4–9.5] | NA | 1 | 919 | 1113 | NA | NA | 1 | NA |
| Smoke Exposure | 3 [1.8–5.2] | [0.4–22.9] | 5 | 2825 | 20871 | 3.1 [2.2–4.5] | 89.8 [79–95] | < 0.001 | 0.467 |
| Syphilis | 7.5 [5.3–10.8] | NA | 1 | 221 | 473 | NA | NA | 1 | NA |
| Thyroid disorder | 1 [0.7–1.3] | NA | 1 | 404 | 359 | NA | NA | 1 | NA |
| Total Bilirubin (mg_dL), Not Normal | 2 [1.4–3] | NA | 2 | 1661 | 6594 | 2.6 [1.3–5.3] | 85.4 [40.9–96.4] | 0.009 | NA |
| Age | 2.3 [1.5–3.6] | NA | 2 | 216 | 917 | 1.3 | 41.1 | 0.193 | NA |
| Body mass index | 7.5 [5.3–10.8] | NA | 1 | 221 | 473 | NA | NA | 1 | NA |
| Dialysis duration | 1.8 [0.8–4.2] | NA | 2 | 120 | 1043 | 2.4 [1.1–4.8] | 82 [23.9–95.7] | 0.018 | NA |
| Hemoglobin | 1.1 [1–1.3] | NA | 1 | 6382 | 6675 | NA | NA | 1 | NA |

depressed cellular immunity, alveolar macrophage dysfunction, low levels of interferon gamma, reduction of interleukin-12, and micronutrient deficiency. We recognize several potential limitations to this review. In addition to the fact that most of the included studies used multiple diagnostic approaches for TB and DM, other diagnostic methods including ICD codes, medical records and self-reported data may be associated with some inaccuracies. Different risk factors have been reported for pulmonary TB compared to extra-pulmonary TB. Very few included studies, however, differentiated pulmonary TB from extrapulmonary TB [85, 86]. Similarly, very few included studies reported information on DM types (1 or 2 and pre-DM or DM) and participant glycaemic control. However, these are conditions that influence susceptibility to TB [87]. Very few included studies reported treatment status for participant for TB. Normalization of glycaemic status has been established for TB patients receiving treatment [88, 89]. This could therefore have been the cause of the misclassification of cases and controls in the included studies. The above limitations would justify the substantial heterogeneity recorded in this meta-analysis. As previously reported [90], very few studies

included in this meta-analysis were from Africa, thus compromising the generalizability of these results globally. It should also be noted that Africa has the highest rate of undiagnosed DM in the world and may therefore have a specific profile of the association between DM and TB [91].

Due to the inclusion of only observational studies in this meta-analysis, a causal link between DM and the risk of TB cannot be suggested. However, the results of this meta-analysis further strengthen the level of evidence for the association between DM and the risk of TB development. We therefore encourage specific studies on the association between DM and TB in the context of Africa. We advocated public health programs to prevent DM such as strengthening education on risk factors for DM and physical activities and sports. Patients with DM only and healthcare professionals should be educated about their increased risk of active or latent TB development. Two-way screening and management programs for DM and TB including latent TB would help reduce the incidence and burden associated with this double epidemic. Interventional studies to demonstrate the causal link between DM and TB are needed in the future. Further research on the biological mechanism by which DM increases the risk of TB are needed.

## Supporting information

**S1 Fig. Funnel chart for publications of the association between diabetes and tuberculosis in cohort studies.**
(PDF)

**S2 Fig. Funnel chart for publications of the association between diabetes and tuberculosis in case control studies.**
(PDF)

**S3 Fig. Funnel chart for publications of the association between diabetes and tuberculosis in cross-sectional studies.**
(PDF)

**S1 Table. Preferred reporting items for systematic reviews and meta-analyses checklist.**
(PDF)

**S2 Table. Search strategy in Pubmed.**
(PDF)

**S3 Table. Items for risk of bias assessment.**
(PDF)

**S4 Table. Main reasons of non-inclusion of eligible studies.**
(PDF)

**S5 Table. Risk of bias assessment.**
(PDF)

**S6 Table. P-value of Khi-2 and Fisher exact tests for qualitative confounding factors.**
(PDF)

**S7 Table. P-value of Student test for quantitative confounding factors.**
(PDF)

**S8 Table. Subgroup analyses of the association between diabetes and tuberculosis.**
(PDF)

## Author Contributions

**Conceptualization:** Joseph Rodrigue Foe-Essomba, Sebastien Kenmoe, Jean Claude Mbanya, Wilfred Fon Mbacham, Sara Eyangoh.

**Data curation:** Joseph Rodrigue Foe-Essomba, Sebastien Kenmoe, Serges Tchatchouang, Jean Thierry Ebogo-Belobo, Donatien Serge Mbaga, Cyprien Kengne-Ndé, Gadji Mahamat, Ginette Irma Kame-Ngasse, Efietngab Atembeh Noura, Chris Andre Mbongue Mikangue, Alfloditte Flore Feudjio, Jean Bosco Taya-Fokou, Sabine Aimee Touangnou-Chamda, Rachel Audrey Nayang-Mundo, Inès Nyebe, Jeannette Nina Magoudjou-Pekam, Jacqueline Félicité Yéngué, Larissa Gertrude Djukouo, Cynthia Paola Demeni Emoh, Hervé Raoul Tazokong, Arnol Bowo-Ngandji, Eric Lontchi-Yimagou.

**Formal analysis:** Sebastien Kenmoe, Cyprien Kengne-Ndé.

**Funding acquisition:** Sebastien Kenmoe.

**Methodology:** Joseph Rodrigue Foe-Essomba, Sebastien Kenmoe, Serges Tchatchouang, Jean Thierry Ebogo-Belobo, Donatien Serge Mbaga, Cyprien Kengne-Ndé, Gadji Mahamat, Ginette Irma Kame-Ngasse, Efietngab Atembeh Noura, Chris Andre Mbongue Mikangue, Alfloditte Flore Feudjio, Jean Bosco Taya-Fokou, Sabine Aimee Touangnou-Chamda, Rachel Audrey Nayang-Mundo, Inès Nyebe, Jeannette Nina Magoudjou-Pekam, Jacqueline Félicité Yéngué, Larissa Gertrude Djukouo, Cynthia Paola Demeni Emoh, Hervé Raoul Tazokong, Arnol Bowo-Ngandji, Eric Lontchi-Yimagou, Afi Leslie Kaiyven, Valerie Flore Donkeng Donfack, Richard Njouom, Jean Claude Mbanya, Wilfred Fon Mbacham, Sara Eyangoh.

**Project administration:** Sebastien Kenmoe, Jean Claude Mbanya, Wilfred Fon Mbacham, Sara Eyangoh.

**Supervision:** Sara Eyangoh.

**Validation:** Sebastien Kenmoe, Serges Tchatchouang, Jean Thierry Ebogo-Belobo, Donatien Serge Mbaga, Gadji Mahamat, Ginette Irma Kame-Ngasse, Efietngab Atembeh Noura, Chris Andre Mbongue Mikangue, Alfloditte Flore Feudjio, Jean Bosco Taya-Fokou, Sabine Aimee Touangnou-Chamda, Rachel Audrey Nayang-Mundo, Inès Nyebe, Jeannette Nina Magoudjou-Pekam, Jacqueline Félicité Yéngué, Larissa Gertrude Djukouo, Cynthia Paola Demeni Emoh, Hervé Raoul Tazokong, Arnol Bowo-Ngandji, Eric Lontchi-Yimagou, Afi Leslie Kaiyven, Valerie Flore Donkeng Donfack, Richard Njouom, Jean Claude Mbanya, Wilfred Fon Mbacham, Sara Eyangoh.

**Writing – original draft:** Joseph Rodrigue Foe-Essomba, Sebastien Kenmoe.

**Writing – review & editing:** Joseph Rodrigue Foe-Essomba, Sebastien Kenmoe, Serges Tchatchouang, Jean Thierry Ebogo-Belobo, Donatien Serge Mbaga, Cyprien Kengne-Ndé, Gadji Mahamat, Ginette Irma Kame-Ngasse, Efietngab Atembeh Noura, Chris Andre Mbongue Mikangue, Alfloditte Flore Feudjio, Jean Bosco Taya-Fokou, Sabine Aimee Touangnou-Chamda, Rachel Audrey Nayang-Mundo, Inès Nyebe, Jeannette Nina Magoudjou-Pekam, Jacqueline Félicité Yéngué, Larissa Gertrude Djukouo, Cynthia Paola Demeni Emoh, Hervé Raoul Tazokong, Arnol Bowo-Ngandji, Eric Lontchi-Yimagou, Afi Leslie Kaiyven, Valerie Flore Donkeng Donfack, Richard Njouom, Jean Claude Mbanya, Wilfred Fon Mbacham, Sara Eyangoh.

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
