## [Decision Letter · Decision Letter 0]

27 Jun 2021

PONE-D-21-16247

Diabetes mellitus and tuberculosis, a systematic review and meta-analysis with sensitivity analysis for studies comparable for confounders

PLOS ONE

Dear Dr. Eyangoh,

Thank you for submitting your manuscript to PLOS ONE. After careful consideration, we feel that it has merit but does not fully meet PLOS ONE’s publication criteria as it currently stands. Therefore, we invite you to submit a revised version of the manuscript that addresses the points raised during the review process.

We look forward to receiving your revised manuscript.

Kind regards,

Antonio Palazón-Bru, PhD

Academic Editor

PLOS ONE

Journal Requirements:

3. Please amend your authorship list in your manuscript file to include author Cyprien Kengne-Ndé

4. Please include captions for ALL your Supporting Information files at the end of your manuscript, and update any in-text citations to match accordingly. Please see our Supporting Information guidelines for more information: http://journals.plos.org/plosone/s/supporting-information.

Reviewers' comments:

Reviewer's Responses to Questions

**Comments to the Author**

1. Is the manuscript technically sound, and do the data support the conclusions?

Reviewer #1: Yes

Reviewer #2: No

2. Has the statistical analysis been performed appropriately and rigorously? 

Reviewer #1: Yes

Reviewer #2: I Don't Know

3. Have the authors made all data underlying the findings in their manuscript fully available?

Reviewer #1: Yes

Reviewer #2: No

4. Is the manuscript presented in an intelligible fashion and written in standard English?

Reviewer #1: Yes

Reviewer #2: Yes

5. Review Comments to the Author

Reviewer #1: Abstract

Line 40: edit 'do not' as 'did not'. Rather than making a general statement it's good to specify the potential confounders which were missed in previous publications but considered in this current study.

Line 53-55: instead of reporting the pooled estimate by study designs which can be discussed in the body, I think the authors have presented results of from subgroup analysis by the potential confounders which they uniquely considered.

Line 57-58: results for heterogeneity have not been presented previously but the statement presented in these lines come all of a sudden.

Line 60-61: the authors have grounded that DM is associated with an increase in TB risk (latent vs active). Why do then recommend further studies given they have consistent evidence? I am a bit confused with the mixed statement provided in the conclusion. Their recommendation should rather focus on investigations of the biological mechanism that DM increases the risk of TB.

Methods

1. Include line numbers starting from the introduction

2. Literature search: include an active link to PROSPERO Registration

3. Include the number of hits last returned in S2 Table.

4. detailed list/or description of list of confounding factors need to be presented here to help the readers judge the additions that this current review did compared to the previous ones.

5. statistical analysis: while there was no clinical trial, it is not important to mention it here

6. For the purpose of clarity and details, it is good if the authors provide a statement on when they say publication bias exists with the use the Egger's test and with the funnel plot

7. I couldn't find the parameters the authors expected to include for which they criticized the previous reviews.

Results

8. Low-income country: describe the standard (or source) used to classify country income levels. The authors mentioned only one study from LICs while several of them. Please check this.

9. In table 1, include details on the type of effect size reported, sample size, population characteristics, effect size, and variables adjusted for the effect size estimated.

10. Table 1: not clear if to what the column label "pairing" refers to.

Discussion

11. Beyond the epidemiological association, to give some depth into the discussion, I suggest the authors provide a statement on how their estimated association can be explained biologically.

Reviewer #2: This study addresses an important topic of high global health importance. The meta-analysis evaluates the association between diabetes and tuberculosis risk. A meta-analysis on the same topic was published in Plos One in 2017 (reference 15 of the manuscript) and included 44 studies, while this review includes 48 studies. The authors claim that it was necessary to reassess tuberculosis risk among patients with diabetes to include a sensitivity analysis balanced for the potential confounders. This approach is interesting, but, the description of confounders and how they were selected is not clear and not well described, and, therefore, the added value of this study is difficult to understand. Indeed, the message of this meta-analysis does not add something different to the review published in 2017 in the same journal.

Comments

Abstract:

The authors should specify what they mean by “wrong design”.

The authors should specify which confounders were accounted for as it is the main difference from previous studies..

Authors present result on TB risk in DM patients but do not specify if it is latent or active TB while in the conclusion, it is stated that DM is associated with an increased risk of active and latent TB.

Main text:

Introduction

Sentence starting with “in 2019, the International…” is not clear.

Introduction could be shorter.

Methods:

What do authors mean by “observational studies at global level”?

Confounders (their selection and criteria lying beyond their selection) should be well described to help the reader to understand the real value of this analysis.

Latent and active TB were not analysed separately (or it is not clear). It does not make sense, from a clinical point of view to group active and latent TB. So, it is important to do separated analysis and draw specific conclusion for each disease stage.

No description of TB and DM diagnosis methods accepted in the selected papers.

6. PLOS authors have the option to publish the peer review history of their article (what does this mean?). If published, this will include your full peer review and any attached files.

Reviewer #1: **Yes: **Melkamu Merid Mengesha

Reviewer #2: No

---

## [Author Response · Author response to Decision Letter 0]

6 Aug 2021

Review Comments to the Author

Reviewer #1: Abstract

Line 40: edit 'do not' as 'did not'. Rather than making a general statement it's good to specify the potential confounders which were missed in previous publications but considered in this current study.

Authors: corrected as proposed, thank you.

Line 53-55: instead of reporting the pooled estimate by study designs which can be discussed in the body, I think the authors have presented results of from subgroup analysis by the potential confounders which they uniquely considered.

Authors: We took into account confounding factors through a sensitivity analysis including only studies with similar proportions of the different confounders between cases and controls. As the range of these confounders is wide, we have reported in the abstract a summary to indicate that the results of this sensitivity analysis were not different from the overall estimate or by study design.

Line 57-58: results for heterogeneity have not been presented previously but the statement presented in these lines come all of a sudden.

Authors: We have now indicated for all estimates the value of I2 (>70%) which indicates substantial heterogeneity.

Line 60-61: the authors have grounded that DM is associated with an increase in TB risk (latent vs active). Why do then recommend further studies given they have consistent evidence? I am a bit confused with the mixed statement provided in the conclusion. Their recommendation should rather focus on investigations of the biological mechanism that DM increases the risk of TB.

Authors: corrected as proposed, thank you.

Methods

1. Include line numbers starting from the introduction

Authors: corrected as proposed, thank you.

2. Literature search: include an active link to PROSPERO Registration

Authors: corrected as proposed, thank you.

3. Include the number of hits last returned in S2 Table.

Authors: corrected as proposed, thank you.

4. detailed list/or description of list of confounding factors need to be presented here to help the readers judge the additions that this current review did compared to the previous ones.

Authors: The list of the main confounding factors has been added in the methodology, data extraction section, thank you.

5. statistical analysis: while there was no clinical trial, it is not important to mention it here

Authors: corrected as proposed, thank you.

6. For the purpose of clarity and details, it is good if the authors provide a statement on when they say publication bias exists with the use the Egger's test and with the funnel plot

Authors: corrected as proposed, thank you.

7. I couldn't find the parameters the authors expected to include for which they criticized the previous reviews.

Authors: We added the list of confounding factors that are unique to this review in the methodology, data extraction section, thank you.

Results

8. Low-income country: describe the standard (or source) used to classify country income levels. The authors mentioned only one study from LICs while several of them. Please check this.

Authors: We have defined the income level of countries according to the World Bank classification, the reference is added in the methodology, data analysis section, thank you. We found only one study from low-income countries (Haraldsdottir, 2015, Guinea-Bissau), but 7 other included articles were from Lower-middle-income economies.

9. In table 1, include details on the type of effect size reported, sample size, population characteristics, effect size, and variables adjusted for the effect size estimated.

Authors: In the present meta-analysis, the methodology approach does not take into account either the effect size reported in the included studies by the primary authors or the adjustment of the analyses for confounding factors. The meta-analysis approach in this our study recalculates all effect estimates from the size of cases, controls and the number with outcome in both groups. All of these numbers and effects are fully presented in Figure 2. We are also re-assessed the confounders from the primary data from the included studies, thank you.

10. Table 1: not clear if to what the column label "pairing" refers to.

Authors: These are matched parameters between cases and controls, we changed the column title in the table accordingly, thank you.

Discussion

11. Beyond the epidemiological association, to give some depth into the discussion, I suggest the authors provide a statement on how their estimated association can be explained biologically.

Authors: been suggested to underlie the association between DM and TB, thank you.

“Little is known about the biological mechanisms that support a high risk of developing TB in patients with DM. Several hypotheses linked to an alteration of immune function in diabetics have however been suggested to explain this association between DM and TB [1-4]. These hypotheses include, but not limited to: depressed cellular immunity, alveolar macrophage dysfunction, low levels of interferon gamma, reduction of interleukin-12, and micronutrient deficiency.”

Reviewer #2: This study addresses an important topic of high global health importance. The meta-analysis evaluates the association between diabetes and tuberculosis risk. A meta-analysis on the same topic was published in Plos One in 2017 (reference 15 of the manuscript) and included 44 studies, while this review includes 48 studies. The authors claim that it was necessary to reassess tuberculosis risk among patients with diabetes to include a sensitivity analysis balanced for the potential confounders. This approach is interesting, but, the description of confounders and how they were selected is not clear and not well described, and, therefore, the added value of this study is difficult to understand. Indeed, the message of this meta-analysis does not add something different to the review published in 2017 in the same journal.

Authors: Thank you for this summary.

Comments

Abstract:

The authors should specify what they mean by “wrong design”.

Authors: We added in the methodology section what we mean by wrong design which includes reviews, case reports, case series… thank you.

The authors should specify which confounders were accounted for as it is the main difference from previous studies.

Authors: We collected all the socio-demographic and clinical confounding factors in the included studies and presented in supplementary tables 6 and 7. We added in the methodology section the major confounders, thank you.

Authors present result on TB risk in DM patients but do not specify if it is latent or active TB while in the conclusion, it is stated that DM is associated with an increased risk of active and latent TB.

Authors: For the included studies, we showed in Table 1 the stage of tuberculosis (variable “TB stage”). In S8 Table, the subgroup analyses performed showed that diabetes mellitus was associated with a risk of active and latent TB.

Main text:

Introduction

Sentence starting with “in 2019, the International…” is not clear.

Authors: The sentence was edited for clarity, thank you.

Introduction could be shorter.

Authors: While we are keen to reduce the length of the introduction, we also feel that with less than 2 pages currently, this introduction does not seem long enough, thank you.

Methods:

What do authors mean by “observational studies at global level”?

Authors: The sentence was edited for clarity, thank you.

Confounders (their selection and criteria lying beyond their selection) should be well described to help the reader to understand the real value of this analysis.

Authors: We further described the confounding factor in the abstract and methodology sections, thank you.

Latent and active TB were not analysed separately (or it is not clear). It does not make sense, from a clinical point of view to group active and latent TB. So, it is important to do separated analysis and draw specific conclusion for each disease stage.

Authors: For the included studies, we showed in Table 1 the stage of tuberculosis (variable “TB stage”). In S8 Table, the subgroup analyses performed showed that diabetes mellitus was associated with a risk of active and latent TB.

No description of TB and DM diagnosis methods accepted in the selected papers.

Authors: For the included studies, we showed in Table 1 the TB and DM diagnosis approaches (variables “TB diagnosis approach” and “DM diagnosis approach”), thank you.

---

## [Decision Letter · Decision Letter 1]

29 Nov 2021

Diabetes mellitus and tuberculosis, a systematic review and meta-analysis with sensitivity analysis for studies comparable for confounders

PONE-D-21-16247R1

Dear Dr. Eyangoh,

We’re pleased to inform you that your manuscript has been judged scientifically suitable for publication and will be formally accepted for publication once it meets all outstanding technical requirements.

Kind regards,

Antonio Palazón-Bru, PhD

Academic Editor

PLOS ONE

Additional Editor Comments (optional):

Reviewers' comments:

Reviewer's Responses to Questions

**Comments to the Author**

1. If the authors have adequately addressed your comments raised in a previous round of review and you feel that this manuscript is now acceptable for publication, you may indicate that here to bypass the “Comments to the Author” section, enter your conflict of interest statement in the “Confidential to Editor” section, and submit your "Accept" recommendation.

Reviewer #1: All comments have been addressed

Reviewer #2: All comments have been addressed

2. Is the manuscript technically sound, and do the data support the conclusions?

Reviewer #1: Yes

Reviewer #2: Yes

3. Has the statistical analysis been performed appropriately and rigorously? 

Reviewer #1: Yes

Reviewer #2: Yes

4. Have the authors made all data underlying the findings in their manuscript fully available?

Reviewer #1: Yes

Reviewer #2: Yes

5. Is the manuscript presented in an intelligible fashion and written in standard English?

Reviewer #1: Yes

Reviewer #2: Yes

6. Review Comments to the Author

Reviewer #1: No further comments; the authors have addressed most of the concerns I raised in the previous submission.

Reviewer #2: (No Response)

7. PLOS authors have the option to publish the peer review history of their article (what does this mean?). If published, this will include your full peer review and any attached files.

Reviewer #1: **Yes: **Melkamu Merid

Reviewer #2: No

---

## [Editor Report · Acceptance letter]

2 Dec 2021

PONE-D-21-16247R1 

Diabetes mellitus and tuberculosis, a systematic review and meta-analysis with sensitivity analysis for studies comparable for confounders. 

Dear Dr. Eyangoh:

I'm pleased to inform you that your manuscript has been deemed suitable for publication in PLOS ONE. Congratulations! Your manuscript is now with our production department. 

Kind regards, 

on behalf of

Dr. Antonio Palazón-Bru 

Academic Editor

PLOS ONE